# Impact of Lactate Clearance on Early Outcomes in Pediatric ECMO Patients

**DOI:** 10.3390/medicina57030284

**Published:** 2021-03-18

**Authors:** Julia Merkle-Storms, Ilija Djordjevic, Carolyn Weber, Soi Avgeridou, Ihor Krasivskyi, Christopher Gaisendrees, Navid Mader, Ferdinand Kuhn-Régnier, Axel Kröner, Gerardus Bennink, Anton Sabashnikov, Uwe Trieschmann, Thorsten Wahlers, Christoph Menzel

**Affiliations:** 1Heart Centre, Department of Cardiothoracic Surgery, University of Cologne, 50924 Cologne, Germany; carolyn.weber@uk-koeln.de (C.W.); soi.avgeridou@uk-koeln.de (S.A.); ihor.krasivskyi@uk-koeln.de (I.K.); christopher.gaisendrees@uk-koeln.de (C.G.); navid.mader@uk-koeln.de (N.M.); ferdinand.kuhn-regnier@uk-koeln.de (F.K.-R.); axel.kroener@uk-koeln.de (A.K.); gerardus.bennink@uk-koeln.de (G.B.); anton.sabashnikov@uk-koeln.de (A.S.); thorsten.wahlers@uk-koeln.de (T.W.); 2Anaesthesiology and Intensive Care Medicine, University of Cologne, 50924 Cologne, Germany; uwe.trieschmann@uk-koeln.de (U.T.); christoph.menzel@uk-koeln.de (C.M.)

**Keywords:** lactate, lactate clearance, ECMO, pediatric ECMO patients, early outcomes, survival

## Abstract

*Background and Objectives:* Pediatric extracorporeal membrane oxygenation (ECMO) support is often the ultimate therapy for neonatal and pediatric patients with congenital heart defects after cardiac surgery. The impact of lactate clearance in pediatric patients during ECMO therapy on outcomes has been analyzed. Materials *and*
*Methods:* We retrospectively analyzed data from 41 pediatric vaECMO patients between January 2006 and December 2016. Blood lactate and lactate clearance have been recorded prior to ECMO implantation and 3, 6, 9 and 12 h after ECMO start. Receiver operating characteristic (ROC) analysis was used to identify cut-off levels for lactate clearance. *Results:* Lactate levels prior to ECMO therapy (9.8 mmol/L vs. 13.5 mmol/L; *p* = 0.07) and peak lactate levels during ECMO support (10.4 mmol/L vs. 14.7 mmol/L; *p* = 0.07) were similar between survivors and nonsurvivors. Areas under the curve (AUC) of lactate clearance at 3, 9 h and 12 h after ECMO start were significantly predictive for mortality (*p* = 0.017, *p* = 0.049 and *p* = 0.006, respectively). Cut-off values of lactate clearance were 3.8%, 51% and 56%. Duration of ECMO support and respiratory ventilation was significantly longer in survivors than in nonsurvivors (*p* = 0.01 and *p* < 0.001, respectively). *Conclusions:* Dynamic recording of lactate clearance after ECMO start is a valuable tool to assess outcomes and effectiveness of ECMO application. Poor lactate clearance during ECMO therapy in pediatric patients is a significant marker for higher mortality.

## 1. Introduction

Currently, approximately 2–3% of all neonatal and pediatric patients undergoing cardiac surgery for congenital heart defect need post-surgical extracorporeal membrane oxygenation (ECMO) support [1]. There are various determination methods for insufficient tissue perfusion, such as measurement of lactate level, mixed venous oxygen saturation (ScVO2), blood pressure, urine output, use of regional near-infrared spectroscopy (NIRS), and inotrope requirement at any time point of ECMO therapy [2]. Nonetheless, the value of these means is quite limited in complex ECMO settings. In addition, only a few previous studies showed that measurement of blood lactate clearance can be proportional to tissue oxygen demand and thus be a reliable prognostic predictor for outcomes [2,3]. High lactate values can predict occurrence of multiorgan failure and inadequate tissue perfusion [3,4,5]. Elevated lactate levels have also been proven to be associated with increased mortality associated with sepsis, trauma, cardiogenic and hemorrhagic shock, regional ischemia, and after surgical procedures including cardiac surgery [5,6]. Lactate, blood pH, partial oxygen and carbon dioxide pressures (paO_2_ and paCO_2_) in arterial blood are easy to obtain using the arterial blood gas analysis (ABG). Thus, over the years lactate clearance has been regarded as a surrogate parameter for the health status of patients and especially for ECMO outcome. Hence, there was only poor information about the inquiry of dynamic lactate clearance levels on adult ECMO patients [3,6]. In addition to this, very little data are available on the impact of hyperlactatemia and lactate clearance for outcomes on pediatric ECMO patients. In this study, we sought to determine whether there is a correlation between lactate clearance in the first 24 h after ECMO start and outcome on neonatal and pediatric patients. Primary outcome was mortality during hospitalization.

## 2. Materials and Methods

This retrospective analysis was performed with 41 pediatric vaECMO patients referred to our center between January 2006 and December 2016. We analyzed neonatal as well as pediatric patients undergoing cardiac surgery for congenital heart defects needing post-surgical ECMO therapy. Among all patients, indication for ECMO application was failure to wean from cardiopulmonary bypass after congenital cardiac surgery (CCS) of 18 patients (43.9%), low cardiac output syndrome (LCOS) with high need of catecholamine following CCS of 7 patients (17.1%), cardiac arrest on intensive care unit of 10 patients (24.4%; 8 of them following CCS), out of hospital cardiac arrest (OHCA) in 4 patients (9.8%), and cardiac arrest in pediatric cardiac catheterization lab of 2 patients (4.9%). Twenty-four patients (58.5%) were males and 17 patients were females (41.5%).

The Ethics Committee of the Medical Faculty of the University of Cologne waived ethics approval and written informed consent for data collection from patients.

Blood samples were taken from an indwelling intra-arterial access and determined with a standard based arterial blood gas analyzer (ABL 800 FLEX Radiometer) that underwent daily calibration and quality-control checks. Blood pH, lactate level, paO2, and paCO2 of arterial blood gas (ABG) analysis prior to ECMO implantation and during the first 24 h on ECMO therapy were recorded. The first arterial blood that contained arterial serum lactate concentration was taken 30 min to 60 min after ECMO start. Furthermore, data of blood serum concentrations of creatinine, aspartate and alanine transaminases (AST, ALT), creatine kinase and creatine kinase MB (CK, CK-MB), C-reactive protein (CRP), bilirubin, international normalized ratio (INR) values, and high sensitive troponin T (hsTnT) prior to ECMO implantation and for 3 days after ECMO implantation were daily collected. Blood tests were assayed by routine automated laboratory techniques (Cobas System 600, Roche Diagnostics GmbH, Mannheim, Germany).

We calculated lactate clearance by lactate concentrations over time ((Lactate_initial_−Lactate_delayed_) × 100%/Lactate_initial_ (mmol/L × hour)). Lactate clearances are given in percent per hour (%/h). Lactate_initial_ is lactate level before ECMO implantation. Lactate_delayed_ is the lactate level measured at the chosen point of time (3, 6, 9, 12 h after ECMO implantation). We categorized lactate clearance as positive when lactate concentration decreased. Blood lactate concentrations and lactate clearances were serially measured prior to ECMO and every 3 h after initiation of ECMO support for at least 24 h.

Tissue microdialysis was not conducted due to its invasiveness and costs.

Main characteristics, criteria and technique of vaECMO assistance in neonatal and pediatric patients have been published in detail in previous studies [7,8,9,10,11]. Target activated clotting time was 160–180 s, whereas activated partial thromboplastin time (aPTT) was maintained by 60–80 s., based on daily controls if there was no active bleeding tendency. Generally, initial ECMO flow was maintained by 150–200 mL/kg/min on newborns and 2.4 L/m^2^/min on older children. Adjustment of epinephrine, dobutamine, and norepinephrine infusions was performed to maintain mean arterial pressure at >50 mmHg throughout ECMO runtime. Ventilation was modulated to a paCO2 of 40 mmHg and a paO2 between 100 and 200 mmHg. Arterial lactate values of 0.5 to 2.0 mmol/L were considered as normal range.

Successful ECMO weaning was defined as survival more than 12 h after ECMO weaning and separation. Patients who survived longer than 12 h were defined as survivors, and patients who did not survive 12 h after ECMO start were defined as nonsurvivors. All-cause mortality was defined as death from any cause. Patients who received ECMO under ongoing conventional cardiopulmonary resuscitation (cCPR) without achieving return of spontaneous circulation (ROSC) were considered as patients with extracorporeal cardiopulmonary resuscitation (eCPR).

In this paper the authors focused on lactate and lactate clearance. Involvement and analysis of other factors such as HCO_3_ were not conducted though these factors might also play an important role.

Between 2006 and 2016, we could only include 41 children meeting our inclusion criteria for the study. Due to the relatively small patient cohort, calculation of the sample size was not constructive. Our results should be confirmed by future studies with a larger number of patients.

### Statistical Methods

Statistics were performed using Student *t*-Test or Mann–Whitney-*U* test, each depending on whether continuous variables are normally distributed or not, and Chi-square test was used for categorical variables. Continuous variables are expressed as mean ± standard deviation (SD) or median with 25th and 75th interquartile ranges (IQR) when appropriate. Fisher exact test was performed when the minimum expected count of cells was <5. Areas under the curve (AUC) of receiver operating characteristic (ROC) analysis and optimal cut-off values (%/h) for lactate clearance were determined. The optimal cut-off values were defined as the values that provided highest sensitivity and specificity. A *p-*value < 0.05 was considered as significant. Statistical analysis was performed using SPSS Version 25.0 (IBM Corp, Chicago, IL, USA).

## 3. Results

Most baseline characteristics were similarly distributed between survivor and nonsurvivor group except for plasma, HCO_3_, lactate, arterial blood pH and base excess (Table 1).

Peak lactate levels throughout the first 24 h of ECMO support were similar in nonsurvivors compared to survivors (14.7 mmol/L versus 10.4 mmol/L; *p* = 0.07). ECMO duration as well as duration of ventilation time was significantly longer in survivor group than in nonsurvivor (*p* = 0.01 and *p* < 0.001, respectively), whereas need for eCPR was similarly distributed between groups (Table 2).

Receiver operating characteristic (ROC) analysis of lactate clearances at 3, 6, 9, and 12 h after ECMO start showed excellent reliability for predicting mortality on ECMO support with significant results (AUC = 0.80, *p =* 0.017 and AUC = 0.72, *p* = 0.059 and AUC = 0.74, *p* = 0.049 and AUC = 0.80, *p* = 0.006, respectively) (Figure 1, Figure 2, Figure 3 and Figure 4). Cut-off values of lactate clearance were 3.8%, 42.6%, 50.7%, and 55.8% at 3, 6, 9, and 12 h, respectively, after ECMO application. Sensitivities and specificities were 85% versus 75%, 80% versus 56%, 77% versus 75% and 83% versus 70%, respectively. Lactate clearance gives the surgeon a valuable argument for termination of ECMO therapy within 12 h after ECMO start.

Laboratory parameters such as creatinine kinase, INR, bilirubin or creatinine prior to ECMO therapy were similar between survivors and nonsurvivors but on 1st, 2nd and 3rd day of ECMO therapy, parameters significantly differed between groups in favor of survivors (Table 3). Table 3 displays the significant increase of the liver parameter AST on 2nd and 3rd day after ECMO implantation when lactate values have already significantly increased leading to acidosis with successive hemodynamic instability, multiorgan failure and final affection of the liver.

## 4. Discussion

ECMO support is claimed to be a very effective tool for maintaining systemic circulation in cardiac arrest [6]. Anaerobic glycolysis and tissue hypoxia caused by cardiac arrest in the majority of adult patients prior to ECMO application forces hyperlactatemia and metabolic acidosis [4]. Li et al. noticed that elevated lactate values following ECMO implantation correlate with higher mortality in adult patients [12]. Hence, downsizing of lactate levels and improvement of lactate clearance may improve perfusion and oxygenation during ECMO therapy. Serial lactate clearance measurements in adult patients have been reported to be a clinically more descriptive term than pure lactate values as a surrogate parameter for the magnitude of global tissue hypoxia [13,14,15]. In addition, Singh et al. revealed a correlation between lactate clearance and survival in adult patients, corroborating our results but in pediatric patients [14].

Generally, lactate contains hydrogen ions that influence the pH, provoking an acidulous pH shift. This acidulous shift and acidulous accumulation finally lead to organ dysfunction. The kidney is the most sensitive organ in the body reflecting severe hemodynamic dysfunction. With decreasing hemodynamic stability, the body produces more lactate that affects to a higher extent kidney function. Therefore, we conclude that increasing lactate levels lead to an amplification of acidosis, especially in nonsurvivors, leading to a decrease of kidney function with consecutive affecting of lactate clearance.

Likewise, parameters such as mixed venous oxygen saturation were commonly used on intensive care units [8]. However, they cannot accurately determine tissue hypoxia. Therefore lactate clearance is an ideal parameter to determine anaerobic conditions [16]. Indeed, lactate clearance in tissues follows different kinetics compared to the lactate clearance in blood. Tissue microdialysis is a good method for measurement of lactate clearance. Microdialysis is a minimally invasive sampling technique that is used for continuous measurement of certain concentrations in the extracellular fluid of tissue. Thus, an assessment of biochemical processes in the body can be achieved. Additionally their distribution within the body can be analyzed. However, microdialysis technique requires the insertion of a small microdialysis catheter (also referred to as microdialysis probe) into the tissue of interest. Due to the retrospective character of our study, we did not use the tissue microdialysis method. In our institution we did not insert further catheters in the treated children and we only analyzed blood probes. Despite scientific advances in making microdialysis probes smaller and more efficient, the invasive nature of this technique still poses some practical and ethical limitations, especially in children. Therefore, we preferred analyzing lactate clearance only in blood. Additionally, tissue microdialysis is also linked with higher costs than only analyzing lactate clearance in blood samples.

To the best of the authors’ knowledge, this study is one of the first assessing lactate clearance as a predictor for mortality in neonatal and pediatric patients in the first 24 h after ECMO start. Only a few studies investigated lactate clearance in adult patients [3,14,15].

Lactate is recognized as a prognostic factor in several critical conditions. In patients with severe hemodynamic instability after congenital surgery, ECMO implantation is a well-established therapy when patients are otherwise unresponsive to conventional therapy and echocardiography. Scolari et al. also only focused on lactate and lactate clearance in their 2020 published paper and found that serum lactate was an important prognostic biomarker in cardiogenic shock treated with ECMO. They also concluded that serum lactate and lactate clearance at 24 h were the strongest independent predictors of short-term survival [1]. Despite lactate and lactate clearance, also other factors, such as HCO_3,_ play an important role. In our study, HCO_3_ was analyzed in survivors versus nonsurvivors prior to ECMO therapy, already revealing a significant tendency for nonsurvivors (Table 1).

In our study, lactate clearances were determined prior to as well as 3, 6, 9 and 12 h after ECMO initiation in neonatal and pediatric patients. A significantly higher lactate clearance was observed on survivors after ECMO start compared to nonsurvivors, correlating with results in adult patients by Slottosch et al. [3]. Zang et al. revealed a cut-off point for lactate clearance of adult patients six hours after vvECMO initiation of 17.5% [17]. Our study revealed a cut-off point of 42.6% in neonatal and pediatric patients with vaECMO therapy.

Elevated lactate levels correlate most likely to a persistent deficit of oxygen delivery or a damaged microcirculation resulting in tissue ischemia. High lactate levels over a brief time period after ECMO start can result from a “washout effect” of underperfused tissues. Furthermore, some previous studies suggested that hyperglycemia might stimulate tissue glucose uptake and glycolysis that contribute to hyperlactatemia [18,19]. These cognitions suggest that lactate levels might be indicative for patients’ global and cardiac status only at the time of cardiogenic shock or a short time after ECMO start. Therefore, serial lactate and lactate clearance measurements after ECMO start are of importance.

Lactate clearance has scarcely been proven to correlate with median arterial pressure (MAP) and inotrope need [20]. Though MAP prior to ECMO start was similar in nonsurvivors and survivors, lactate clearance was higher in survivors after decline of initial cardiogenic shock. Therefore lactate clearance is a more reliable parameter than MAP.

Increasing lactate values indicate hemodynamic instability due to cardiogenic shock or other reasons. This leads to acidosis and severe impairment of organs with rising lactate levels. The kidney is one of the most sensitive organs in the body for reflecting pathophysiologic mechanisms. As many studies have already presented, development of an acute kidney injury is a strong predictor for increased mortality. Therefore, it is consequent that with increasing lactate values the kidney function decreases. This did not affect the result, but explains a pathologic mechanism.

In a relatively large study cohort of adult patients, Park et al. showed that monitoring of lactate values can be used as a reliable tool for monitoring adequate tissue perfusion during extracorporeal life support and that this is strongly predictive of mortality [21]. Buijs et al. described that arterial lactate values predict mortality in pediatric patients who underwent ECMO application, but they did not analyze lactate clearances [22,23], and only children with severe respiratory failure with need for ECMO therapy were enrolled. Primary cardiac failure indications were excluded from the study.

One of the weaknesses was the nonrandomized nature of the study. Due to the fact that studies with children are generally more complex and need more careful study preparation than studies with adults, we believe that dealing with 41 children we can already present sufficient results, and of course in the future bigger studies dealing with more patients are needed. Moreover, in this study the focus was on lactate as well as on lactate clearance, and other important factors, such as HCO_3,_ have not been considered. We have already published other papers dealing with other risk factors such as predictors for survival (see references 9 and 10). Furthermore, one should keep in mind that the primarily used method was analysis of blood samples, though tissue microdialysis would also be a valuable method for analyzing lactate and lactate clearance.

Nevertheless, analyzing data of a cohort of 41 children is more than many other studies dealing with children present. Another strength of our paper is detailed pre-ECMO parameter presentation and the exact analysis of lactate parameters and lactate clearances over a period of time at 3, 6, 9, and 12 h after ECMO implantation. Additionally, the main strength is the establishment of lactate cut-off values via ROC analysis.

## 5. Conclusions

Lactate clearance is an excellent marker for the assessment of ECMO therapy in neonatal and pediatric patients after cardiac surgery. Serial measurement of lactate clearance in the first 24 h after ECMO start is the strength of the paper and could help to identify patients with high risk for morbidity and mortality. Poor lactate clearances correlated with higher morbidity and mortality. Cut-off values of lactate clearances, established in our study, can help to assess outcome.

## Figures and Tables

**Figure 1 medicina-57-00284-f001:**
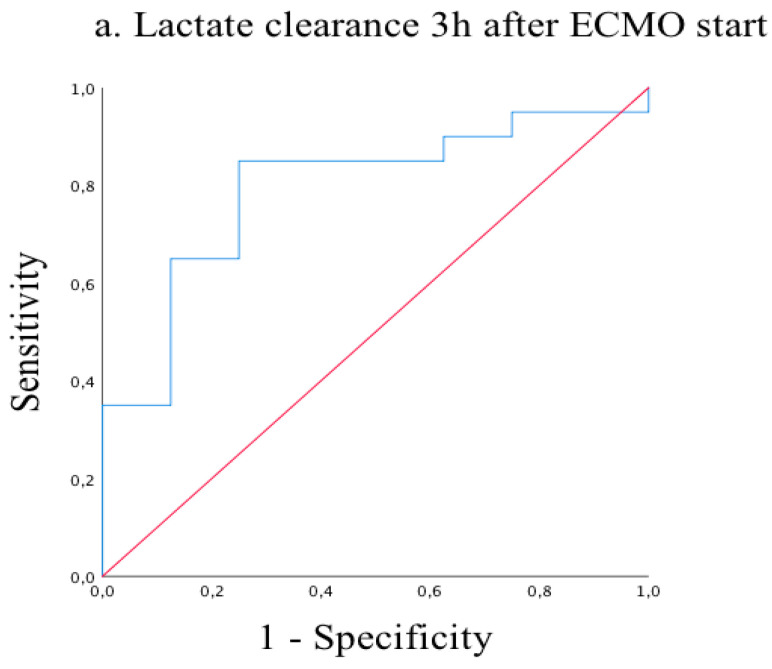
Receiver operating characteristic (ROC) curve of lactate clearance 3 h after ECMO start. AUC = 0.80 (95% CI 0.61–0.97; *p* = 0.017).

**Figure 2 medicina-57-00284-f002:**
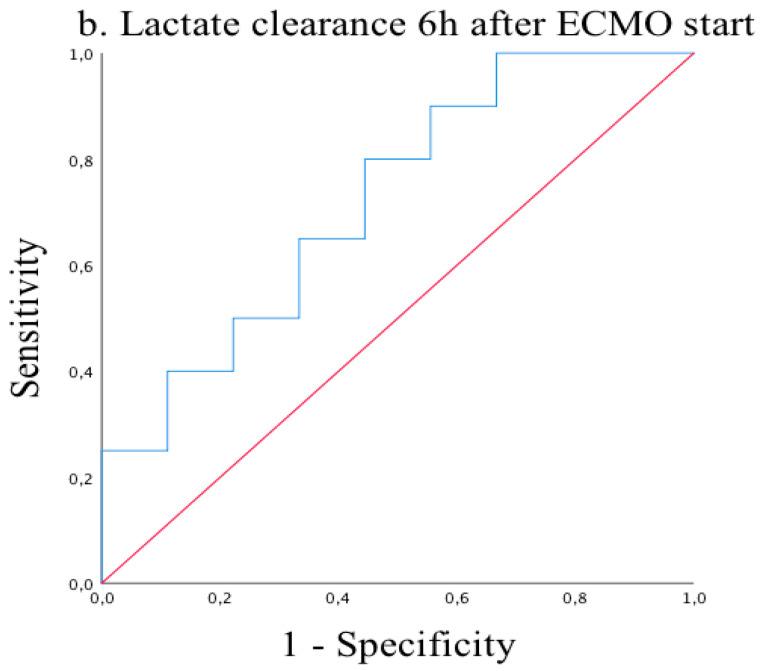
Receiver operating characteristic (ROC) curve of lactate clearance 6 h after ECMO start. AUC = 0.72 (95% CI 0.51–0.92; *p* = 0.059).

**Figure 3 medicina-57-00284-f003:**
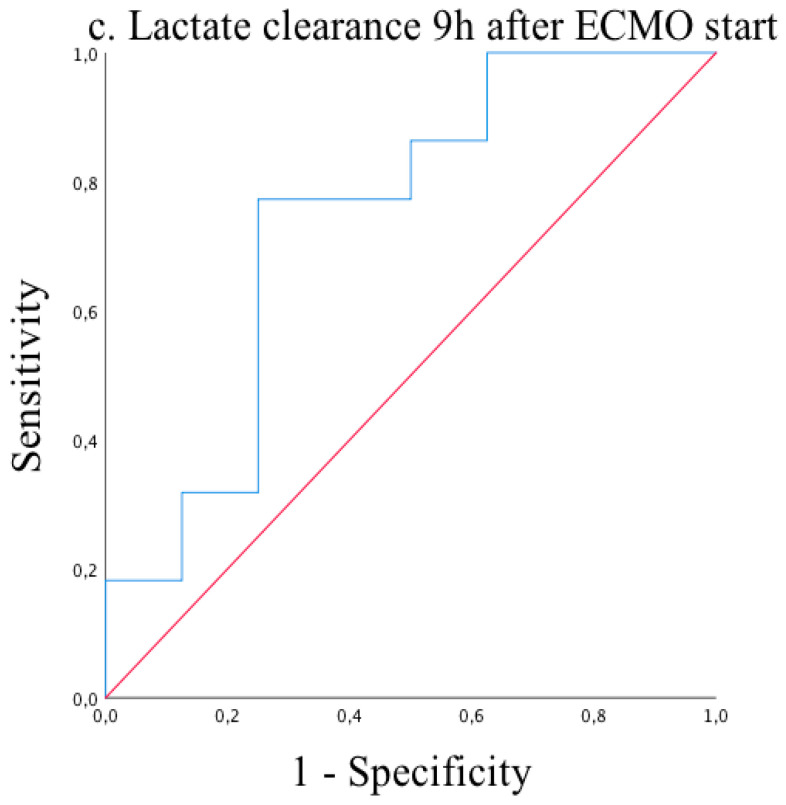
Receiver operating characteristic (ROC) curve of lactate clearance 9 h after ECMO start. AUC = 0.74 (95% CI 0.51–0.96; *p* = 0.049).

**Figure 4 medicina-57-00284-f004:**
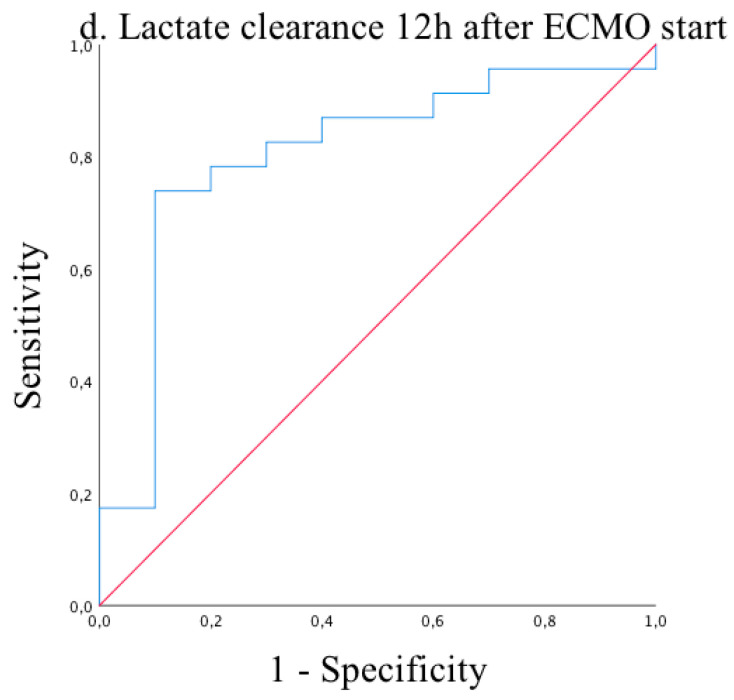
Receiver operating characteristic (ROC) curve of lactate clearance 12 h after ECMO start. AUC = 0.80 (95% CI 0.63–0.97; *p* = 0.006.

**Table 1 medicina-57-00284-t001:** Baseline characteristics prior to ECMO therapy.

	Overall	Survivors	Non-Survivors	*p*-Value
Age (days)	98(12;653)	128(16;188)	84(9;5722)	0.77
Gender (m/f)	24/17	18/9	6/8	0.19
Weight (gram)	5040(3247;11,000)	5200(3230;9000)	4600(3050;68,000)	0.44
Length (cm)	58(51;84)	58(51;70)	90(51;180)	0.38
MAP (mmHg)	37(26;43)	35(23;50)	40(27;43)	0.74
Heart rate (min^−1^)	147(131;177)	145(135;180)	160(65;177)	0.95
Blood sugar (mg/dL)	200 ± 86	196 ± 75	209 ± 108	0.97
HCO_3_^−1^ (mEq/L)	20.2 ± 6.1	22.4 ± 5.3	16.8 ± 6.1	0.03
Body temperature (°C)	35.6(34.1;36)	35.8(35.4;36.3)	34(32;36)	0.13
Lactate (mmol/L)	11 ± 5.8	9.8 ± 5.3	13.5 ± 6.1	0.07
Arterial blood pH	7.220(7.075;7.304)	7.273(7.108;7.355)	7.105(6.815;7.265)	0.08
Base excess (mEq/L)	−5.6(−14.7;3.1)	−3.1(−5.6;4.0)	−15(−17;−9)	0.01
paO_2_ (mmHg)	41(32;58)	43(37;57)	41(26;70)	0.50
paCO_2_ (mmHg)	46(35;64)	45(36;61)	50(31;94)	0.55

MAP, mean arterial pressure; ECMO, extracorporeal membrane oxygenation; paO_2_, partial pressure of oxygen in arterial blood; paCO_2_, partial pressure of carbon dioxide in arterial blood; HCO_3_-, hydrogen carbonate.

**Table 2 medicina-57-00284-t002:** Postoperative variables.

	Overall	Survivors	Non-Survivors	*p*-Value
Peak lactate level in the first 24 h of ECMO support (mmol/L)	11.8 ± 6.8	10.4 ± 6.0	14.7 ± 7.1	0.07
Lactate (mmol/L)				
3 h after ECMO start	8.5 ± 5.5	6.9 ± 5.0	10.1 ± 7.3	0.059
6 h after ECMO start	5.3 ± 4.9	3.8 ± 2.7	6.7 ± 6.0	0.015
9 h after ECMO start	4.2 ± 3.0	3.3 ± 2.0	4.4 ± 2.4	0.075
12 h after ECMO start	4.1 ± 3.1	2.9 ± 1.5	4.3 ± 2.2	0.022
ECMO duration (d)	4(2;6)	4(3;7)	1.5(1;3.7)	0.01
Duration of mechanical ventilation (h)	205(78;545)	354(195;665)	55(32;123)	<0.001
eCPR	16(39%)	8(29.6%)	8(57.1%)	0.10

eCPR, extracorporeal cardio-pulmonary resuscitation; ECMO, extracorporeal membrane oxygenation.

**Table 3 medicina-57-00284-t003:** *p*-values of laboratory parameters comparing survivor versus nonsurvivor group during ECMO support. Poorer values showed nonsurvivor group.

	pre-ECMO	1st Day	2nd Day	3rd Day
Creatine kinase	0.72	0.25	0.004	0.001
Creatine kinase MB	0.71	0.01	0.001	<0.001
hsTnT	0.61	0.007	0.001	0.002
INR	0.80	0.63	0.48	0.98
AST (GOT)	0.35	0.17	0.004	0.002
ALT (GPT)	0.49	0.46	0.057	0.068
Bilirubine	0.069	0.63	0.33	0.45
Creatinine	0.38	0.055	0.007	0.002
Creatinine	0.38	0.055	0.007	0.002

MB, muscle brain; INR, international normalized ratio; AST, aspartate-aminotransferase; ALT, alanine aminotransferase.

## Data Availability

On demand at the author’s institution.

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
