# Peer review of "Impact of Lactate Clearance on Early Outcomes in Pediatric ECMO Patients"

_medicina, 2021, doi:10.3390/medicina57030284_

Round 1

Reviewer 1 Report

This is an Interesting and well presented manuscript.

Comment one: Can the authors kindly explain how the effect of substantially higher levels of acidosis in non-survivors could have affected the lactate clearance in non-survivors?

Comment two:  Could the authors be kind enough to explain the two different parameters in Lactate clearance formula? Initial and delayed values were defined based to which point of time? Why the specific time points were chosen?

Comment three: The lactate clearance in tissues follows different kinetics compared to the lactate clearance in blood. Have the authors considered to test their hypothesis with application of tissue lactate clearance measurement methods like tissue microdialysis? Can the authors please answer why not tested the tissue lactate levels for correlation?

Comment four: The lactate clearance in blood is not only depended on the rate of lactate production by the affected tissues but also on the efficacy of the lactate clearance by the liver. How the authors ensured that the efficacy of hepatic lactate clearance has been considered during the analysis of their results? Could the higher levels of AST in non-survivors imply a reduced hepatic reserve for lactate clearance due to metabolic compromise of the liver. INR and ALT and bilirubin are not ideal indicators of hepatic metabolic reserve.

Comment five: Could the authors comment on the potential effect of the significantly more severe renal dysfunction in non-survivors? Could that have affected their results? How did they deal with this potential confounding factor in the analysis?

Comment six: Could the authors please be kind enough to highlight the strengths and weaknesses of their study in the discussion section? It would be useful to compare their study methodologically to the other lactate clearance studies in paediatric and adult population.

Comment seven: What does the MAD abbreviation mean in line 194? Could the authors please explain that abbreviation?

Thank you  

Author Response

Reviewer response to Journal of Medicina Manuscript ID medicina-1122801

The Authors are very grateful to the Editor and the Reviewer involved in the peer review process for their time and effort. All comments were addressed point by point and appropriate changes were performed in the manuscript.

Reviewer comment 1: This is an Interesting and well presented manuscript. Can the authors kindly explain how the effect of substantially higher levels of acidosis in non-survivors could have affected the lactate clearance in non-survivors?

Author response 1: Thank you very much for your interest in our paper. We kindly answer all your comments. Generally, lactate contains hydrogen ions which influence the pH provoking an acidulous pH shift. This acidulous shift and acidulous accumulation finally lead to organ dysfunction. The kidney is the most sensitive organ in the body reflecting severe hemodynamic dysfunction. With decreasing hemodynamic stability the body produces more lactate that affects kidney function to a higher extent. Therefore, we conclude that increasing lactate levels lead to an amplification of acidosis especially in non-survivors leading to a decrease of kidney function with consecutive affection of lactate clearance.

Changes to the manuscript: Section Discussion: Generally, lactate contains hydrogen ions which influence the pH provoking an acidulous pH shift. This acidulous shift and acidulous accumulation finally lead to organ dysfunction. The kidney is the most sensitive organ in the body reflecting severe hemodynamic dysfunction. With decreasing hemodynamic stability the body produces more lactate that affects kidney function to a higher extent. Therefore, we conclude that increasing lactate levels lead to an amplification of acidosis especially in non-survivors leading to a decrease of kidney function with consecutive affection of lactate clearance.

Reviewer comment 2: Could the authors be kind enough to explain the two different parameters in Lactate clearance formula? Initial and delayed values were defined based to which point of time? Why the specific time points were chosen?

Author response 2: This is an important comment. We calculated lactate clearance by lactate concentrations over time ([Lactateinitial−Lactatedelayed] x 100% / Lactateinitial [mmol/L*hour]). Lactate clearances are given in percent per hour (%/h). Lactateinitial  is lactate level before ECMO implantation. Lactatedelayed is the lactate level measured at the chosen point of time (3, 6, 9, 12 hours after ECMO implantation). We categorized lactate clearance as positive when lactate concentration decreased. Blood lactate concentrations and lactate clearances were serially measured prior to ECMO and every 3 hours after initiation of ECMO support for at least 24 hours.

Changes to the manuscript: Section Method: We calculated lactate clearance by lactate concentrations over time ([Lactateinitial−Lactatedelayed] x 100% / Lactateinitial [mmol/L*hour]). Lactate clearances are given in percent per hour (%/h). Lactateinitial  is lactate level before ECMO implantation. Lactatedelayed is the lactate level measured at the chosen point of time (3, 6, 9, 12 hours after ECMO implantation). We categorized lactate clearance as positive when lactate concentration decreased. Blood lactate concentrations and lactate clearances were serially measured prior to ECMO and every 3 hours after initiation of ECMO support for at least 24 hours.

Reviewer comment 3: The lactate clearance in tissues follows different kinetics compared to the lactate clearance in blood. Have the authors considered to test their hypothesis with application of tissue lactate clearance measurement methods like tissue microdialysis? Can the authors please answer why not tested the tissue lactate levels for correlation?

Author response 3: Thank you very much for this valuable comment. Indeed lactate clearance in tissues follows different kinetics compared to the lactate clearance in blood. Tissue microdialysis is a good method for measurement of lactate clearance. Microdialysis is a minimally-invasive sampling technique that is used for continuous measurement of certain concentrations in the extracellular fluid of tissues. Thus an assessment of the biochemical processes in the body can be achieved. Additionally their distribution within the body can be analyzed. However, microdialysis technique requires the insertion of a small microdialysis catheter (also referred to as microdialysis probe) into the tissue of interest. Due to the retrospective character of our study we did not use tissue microdialysis method. In our institution we did not insert further catheters in the treated children and we only analyzed blood probes. Despite scientific advances in making microdialysis probes smaller and more efficient the invasive nature of this technique still poses some practical and ethical limitations especially in children. Therefore we preferred analyzing lactate clearance only in blood. Additionally, tissue microdialysis is also linked with higher costs than only analyzing lactate clearance in blood samples.

Changes to the manuscript 3:

In section Methods: Tissue microdialysis was not conducted due to its invasiveness and costs.

Section discussion: Indeed lactate clearance in tissues follows different kinetics compared to the lactate clearance in blood. Tissue microdialysis is a good method for measurement of lactate clearance.Microdialysis is a minimally-invasive sampling technique that is used for continuous measurement of certain concentrations in the extracellular fluid of tissue. Thus an assessment of biochemical processes in the body can be achieved. Additionally their distribution within the body can be analyzed. However, microdialysis technique requires the insertion of a small microdialysis catheter (also referred to as microdialysis probe) into the tissue of interest. Due to the retrospective character of our study we did not use tissue microdialysis method. In our institution we did not insert further catheters in the treated children and we only analyzed blood probes. Despite scientific advances in making microdialysis probes smaller and more efficient the invasive nature of this technique still poses some practical and ethical limitations especially in children. Therefore we preferred analyzing lactate clearance only in blood. Additionally, tissue microdialysis is also linked with higher costs than only analyzing lactate clearance in blood samples.

Reviewer comment 4: The lactate clearance in blood is not only depended on the rate of lactate production by the affected tissues but also on the efficacy of the lactate clearance by the liver. How the authors ensured that the efficacy of hepatic lactate clearance has been considered during the analysis of their results? Could the higher levels of AST in non-survivors imply a reduced hepatic reserve for lactate clearance due to metabolic compromise of the liver. INR and ALT and bilirubin are not ideal indicators of hepatic metabolic reserve.

Author response 4: Thank you very much for this comment. You are absolutely right that lactate clearance in blood not only depends on the rate of lactate production by the affected tissues but also on the efficacy of lactate clearance by the liver. We also analyzed liver parameters as presented in Table 3. On first day after ECMO implantation liver parameters were similar with the pre-ECMO liver parameters. Not till second day after ECMO implantation with already significant lactate increase liver parameters also decreased. Additionally, we only included children with preoperative normal liver parameters. As described in Table 3 the liver parameter AST significantly increased on second day after ECMO implantation when lactate values have already significantly increased leading to acidosis with successive hemodynamic instability, multiorgan failure and final affection of the liver. Therefore the elevated AST parameter indeed displays impairment of the liver.

Changes to the manuscript 4: Section results: Table 3 displays the significant increase of the liver parameter AST on second day after ECMO implantation when lactate values have already significantly increased leading to acidosis with successive hemodynamic instability, multiorgan failure and final affection of the liver.

Reviewer comment 5: Could the authors comment on the potential effect of the significantly more severe renal dysfunction in non-survivors? Could that have affected their results? How did they deal with this potential confounding factor in the analysis?

Author response 5: Increasing lactate values indicate hemodynamic instability due cardiogenic shock or other reasons. This leads to acidosis and severe impairment of organs with rising lactate values. The kidney is one of the most sensitive organs in the body always displaying that things go wrong. As many studies have already presented development of an acute kidney injury is a strong predictor for increased mortality. Therefore it is consequent that with increasing lactate values the kidney function decreases. This did not affect the result, but explains an pathologic mechanism.

Changes to the manuscript 5: Section discussion: Increasing lactate values indicate hemodynamic instabilitydue cardiogenic shock or other reasons. This leads to acidosis and severe impairment of organs with rising lactate levels. The kidney is one of the most sensitive organs in the body always pathophysiologic mechanisms. As many studies have already presented development of an acute kidney injury is a strong predictor for increased mortality. Therefore it is consequent that with increasing lactate values the kidney function decreases. This did not affect the result, but explains an pathologic mechanism.

Reviewer comment 6: Could the authors please be kind enough to highlight the strengths and weaknesses of their study in the discussion section? It would be useful to compare their study methodologically to the other lactate clearance studies in pediatric and adult population.

Author response 6: Thank you very much for this important comment. So working out our response to the reviewers, strengths and weaknesses of our study become more evident for the reader. We improved several paragraphs within the manuscript: One of the weaknesses is the non-randomized nature of the study. Due to the fact that studies with children are generally more complex and need more careful study preparation than studies with adults we believe that dealing with 41 children we can already present sufficient results and of course in future bigger studies dealing with more patients are needed. Moreover, in this study focus was on lactate as well as on lactate clearance and other important factors, such as HCO3 have not been considered. We have already published other papers dealing with other risk factors as predictors for survival (see references 9 and 10). Furthermore, one should keep in mind that the primarily used method was analysis of blood samples though tissue microdialysis would also be a valuable method for analyzing lactate and lactate clearance.

Nevertheless analyzing data of a cohort of 41 children is more than many other studies dealing with children present. Another strength of our paper is detailed pre-ECMO parameter presentation and the exact analysis of lactate parameters and lactate clearances over a period of time at 3, 6, 9, and 12 hours after ECMO implantation. Additionally the main strength is the establishment of lactate cut-off values via ROC analysis.

Changes to the manuscript 6:

Section discussion: One of the weaknesses was the non-randomized nature of the study. Due to the fact that studies with children are generally more complex and need more careful study preparation than studies with adults we believe that dealing with 41 children we can already present sufficient results and of course in future bigger studies dealing with more patients are needed. Moreover, in this study focus was on lactate as well as on lactate clearance and other important factors, such as HCO3 have not been considered. We have already published other papers dealing with other risk factors as predictors for survivals (see references 9 and 10). Furthermore, one should keep in mind that the primarily used method was analysis of blood samples though tissue microdialysis would also be a valuable method for analyzing lactate and lactate clearance.

Nevertheless analyzing data of a cohort of 41 children is more than many other studies dealing with children present. Another strength of our paper is detailed pre-ECMO parameter presentation and the exact analysis of lactate parameter and lactate clearance over a period of time at 3, 6, 9, and 12 hours after ECMO implantation. Additionally the main strength is the establishment of lactate cut-off values via ROC analysis.

Changes to the manuscript 6:

Section conclusion: Lactate clearance is an excellent marker for the assessment of ECMO therapy in neonatal and pediatric patients after cardiac surgery. Serial measurement of lactate clearance in the first 24 hours after ECMO start is the strength of the paper and could help to identify patients with high risk for morbidity and mortality. Poor lactate clearances correlated with higher morbidity and mortality. Cut-off values of lactate clearances, established in our study, can help to assess outcome.

Reviewer comment 7: What does the MAD abbreviation mean in line 194? Could the authors please explain that abbreviation?

Author response 7: Instead of MAD we meant MAP standing for mean arterial pressure.

Author response 7: See changes in line 194 and 197.

Reviewer 2 Report

The authors described the importance of dynamic changes of lactate during ECMO to expect the prognosis of the patients.

The objective and method is appropriate. However, excluding other factors which could be relating is necessary to come to this conclusion.

There are several significant factors other than Lactate expecially HCO3!!, which might be the good predictor of survivors. 

The author should show the AUC analysis of all the other significant factors to show that Lac is the best predictior.

Author Response

Reviewer response to Journal of Medicina Manuscript ID medicina-1122801

The Authors are very grateful to the Editor and the Reviewers involved in the peer review process for their time and effort. All comments were addressed point by point and appropriate changes were performed in the manuscript.

Reviewer comment 1: The authors described the importance of dynamic changes of lactate during ECMO to expect the prognosis of the patients. The objective and method is appropriate. However, excluding other factors which could be relating is necessary to come to this conclusion. There are several significant factors other than Lactate expecially HCO3!!, which might be the good predictor of survivors. The author should show the AUC analysis of all the other significant factors to show that Lac is the best predictior.

Author response 1: Thank you very for your valuable comment. Indeed, you are absolutely right that several further significant factors might be good predictors for survivors. In this analysis we primarily focused on lactate. We have already published papers dealing with other risk factors as predictors for survivals as you can see in references 9 and 10. In literature other studies also confirm that lactate is a powerful predictor. Slottosch et al. (refence 3) primarily focused on lactate and lactate clearance in adults. We think that lactate is one of the strongest predictors for survival.

Changes to the manuscript:

Section methods: In this paper the authors focused on lactate and lactate clearance. Involvement and analysis of other factors such as HCO3 were not conducted though these factors might also play in important role.

Section discussion: Lactate has been recognized as a prognostic factor in several critical conditions. In patients with severe hemodynamic instability after congenital surgery ECMO implantation is a well-established therapy when patients are otherwise unresponsive to conventional therapy and echocardiography. Scolari et al. also only focused on lactate and lactate clearance in his 2020 published paper and found out that serum lactate was an important prognostic biomarker in cardiogenic shock treated with ECMO. He also concluded that serum lactate and lactate clearance at 24 h were the strongest independent predictors of short-term survival [1]. Despite lactate and lactate clearance also other factors, such as HCO3 play an important role. In our study HCO3 was analyzed in survivors versus non-survivors prior to ECMO therapy already revealing a significant tendency for non-survivors (Table 1).

  1. Scolari, F.L., et al., Association between serum lactate levels and mortality in patients with cardiogenic shock receiving mechanical circulatory support: a multicenter retrospective cohort study. BMC Cardiovasc Disord, 2020. 20(1): p. 496.

Round 2

Reviewer 1 Report

I would like to thank the authors for their response to my comments and the changes in the manuscript

Line 287: the words "an excellent marker" is suggested to be changed to: "a useful marker". In my opinion excellent markers for monitoring and assessing the short and long term effects of invasive procedures do not really exist as all the markers have innate advantages and disadvantages and limitations that would be useful to be highlighted in each manuscript evaluating those markers.    

Reviewer 2 Report

I understand what the authors aimed for. And the manuscript was extensively revised.